# Identifying minimal risk factors for adolescent suicidal ideation and suicide attempts: A machine learning-optimized approach

Catherine Park [1,2], Beom-Chan Lee [3,4]*

1 Division of Digital Healthcare, Yonsei University, Wonju, South Korea, 2 Center for Planetary Health Digital Healthcare, Institute for Planetary Health, Yonsei University, Wonju, South Korea, 3 Department of Health and Human Performance, University of Houston, Houston, Texas, United States of America, 4 Center for Neuromotor and Biomechanics Research, University of Houston, Houston, Texas, United States of America

* blee24@central.uh.edu

## Abstract

This study aimed to develop and validate a machine learning (ML) model to identify the minimal risk factors for adolescent suicidal behaviors, including suicidal ideation and suicide attempts. Data from the Korea Youth Risk Behavior Web-based Survey (2022–2023), including 90,813 adolescents aged 12–18 years, were analyzed. Using multidimensional risk factors spanning sociodemographic, physical and mental health, and behavioral domains, we applied a Random Forest model combined with recursive feature elimination to identify a minimal subset of risk factors (optimal features). Model performance for identifying suicidal ideation and predicting suicide attempts was evaluated via area under the curve (AUC), sensitivity, specificity, and accuracy metrics across the validation datasets. Sadness, loneliness, anxiety, and stress were identified as optimal features, achieving a high AUC, sensitivity, specificity, and accuracy in identifying suicidal ideation and predicting suicide attempts. Additional factors further improved the ML model's predictive performance for suicide attempts, achieving an AUC of 97.28%, sensitivity of 93.49%, specificity of 90.21%, accuracy of 91.85%, a PPV of 90.52%, and an NPV of 93.26%. This study demonstrated the efficacy of ML-driven approaches in identifying critical risk factors for adolescent suicidal behaviors. The findings highlight the potential of ML frameworks to transform suicide prevention strategies and improve mental health outcomes in adolescents.

## Introduction

Adolescent suicide is a global public health concern. According to the World Health Organization, more than 720,000 individuals die by suicide annually, comprising a substantial proportion of adolescents and young adults [1]. Although suicide is the

**Data availability statement:** The data used in this study were obtained from the Korea Disease Control and Prevention Agency through the Korea Youth Risk Behavior Web-based Survey (KYRBS). The datasets are publicly available to all researchers via the official KDCA portal (https://kdca.go.kr/yhs/yhs/main.do). To replicate the study findings, interested researchers may download the raw datasets for the relevant survey years (e.g., 2022 and 2023) in SPSS format. The authors had no special access privileges to these data.

**Funding:** This research was supported by the Ministry of Science and ICT (MSIT), Korea, under the National Program in Medical AI Semiconductor, supervised by the Institute of Information & Communications Technology Planning & Evaluation (IITP) in 2026 (2024-0-00096 to C.P.), and by the Regional Innovation System & Education (RISE) program through the Gangwon RISE Center, funded by the Ministry of Education and Gangwon State, Republic of Korea (2026-RISE-10-006 to C.P.).

**Competing interests:** The authors have declared that no competing interests exist.

third leading cause of death among adolescents worldwide [1], the prevalence of suicidal ideation and behavior among adolescents varies considerably across regions and countries. A study analyzing data from 90 countries reported a high prevalence of suicidal thoughts among adolescents aged 13–17 years [2]. A recent study revealed that while suicide rates among individuals aged 10–24 years generally declined between 1990 and 2021, some regions continue to report alarmingly high rates [3].

The multifaceted nature of adolescent suicides requires a comprehensive understanding of its underlying determinants. Psychological factors, such as depression and anxiety, have long been recognized as primary contributors to adolescent suicide [4,5]. However, recent umbrella reviews and meta-analyses have highlighted the important role of sociodemographic factors (e.g., academic achievement, academic stress, family structure, and family socioeconomic status) and health-related lifestyle factors (e.g., physical activity, smoking, and alcohol consumption) in contributing to suicidal behaviors among adolescents [6–8]. This growing body of evidence underscores the need for methodologies capable of analyzing and integrating these diverse and complex risk factors.

In recent years, machine learning (ML) has revolutionized suicide risk assessments in adolescents by leveraging large-scale electronic health records and survey data to identify the intricate patterns and interactions among risk factors. Recent systematic reviews and meta-analyses have indicated the increasing application of ML models with algorithms such as random forest and extreme gradient boosting to identify and predict suicidal behaviors (i.e., suicidal ideation and suicide attempts) [9,10]. These studies identified a broad spectrum of risk factors, including psychological determinants, social and familial influences, and behavioral and lifestyle attributes [9,10]. Notably, ML-based approaches have outperformed conventional statistical methods, effectively handling high-dimensional data, capturing nonlinear and intricate patterns, and adapting dynamically to diverse data sources [9,10]. Furthermore, their scalability and flexibility enable early detection, enhanced predictive accuracy, and personalized interventions [9,10].

Despite these advancements, the minimal subset of risk factors required for the accurate and reliable identification and prediction of suicidal ideation and suicide attempts is yet to be established. This could potentially reduce computational overheads, enhance the interpretability of ML models, and facilitate clinical integration. To address this gap, this study aimed to 1) develop and validate an ML model that incorporates multidimensional risk factors, including sociodemographic characteristics, physical and mental health, and health-related behaviors, to identify and predict suicidal ideation and suicide attempts and 2) determine the minimal subset of risk factors (i.e., optimal features) required to ensure accurate, reliable, and clinically actionable identification of suicidal ideation and prediction of suicide attempts.

## Methods

### Data source, participant selection, and multidimensional risk factors

We utilized datasets from the Korea Youth Risk Behavior Web-based Survey (KYRBS) collected from 2022 to 2023. The KYRBS is a nationwide, self-administered

survey designed to monitor health behaviors and risk factors among Korean adolescents (aged 12–18 years) and has been conducted annually by the Korea Disease Control and Prevention Agency since 2005 [11]. The 2022 and 2023 KYRBS were conducted as independent, nationally representative cross-sectional surveys using random sampling of schools and students in South Korea; based on direct confirmation from the KYRBS administrators, there was no respondent overlap between the two survey years.

The KYRBS survey consists of 15 independent sections, including 103 questions that address sociodemographic characteristics, physical and mental health, and health-related behaviors. The use of the KYRBS data and study protocol were approved by the Institutional Review Board of Yonsei University (IRB No. 1041849–202411-BM-226–01; approval date: November 7, 2024), and the study was conducted in accordance with the Declaration of Helsinki. Data access for research purposes occurred on November 8, 2024, following IRB approval. Because this retrospective analysis relied exclusively on anonymized data, the need for written informed consent was waived in accordance with national regulations and institutional policies.

Fig 1 illustrates the participant selection and group classification processes. We initially extracted responses from 54,383 adolescents from the 2022 KYRBS and 55,413 adolescents from the 2023 KYRBS, totaling 109,796 participants. Adolescents who did not respond to any of the 103 questions were excluded from the analysis (N = 18,983), resulting in a final sample size of 90,813 adolescents included in this study. A total of 90,813 adolescents were classified into two groups based on their responses to questions regarding suicidal thoughts and suicide planning as well as well-established classification criteria [3]. The non-suicidal ideation group included adolescents who provided negative (or No) responses to both suicidal thoughts and suicide planning (n = 78,975). The suicidal ideation group included adolescents who provided positive (Yes) responses to suicidal thoughts, suicide planning, or both (n = 11,838). Suicidal ideation was assessed by the question asking whether the participant had ever seriously considered suicide during the past 12 months (yes/

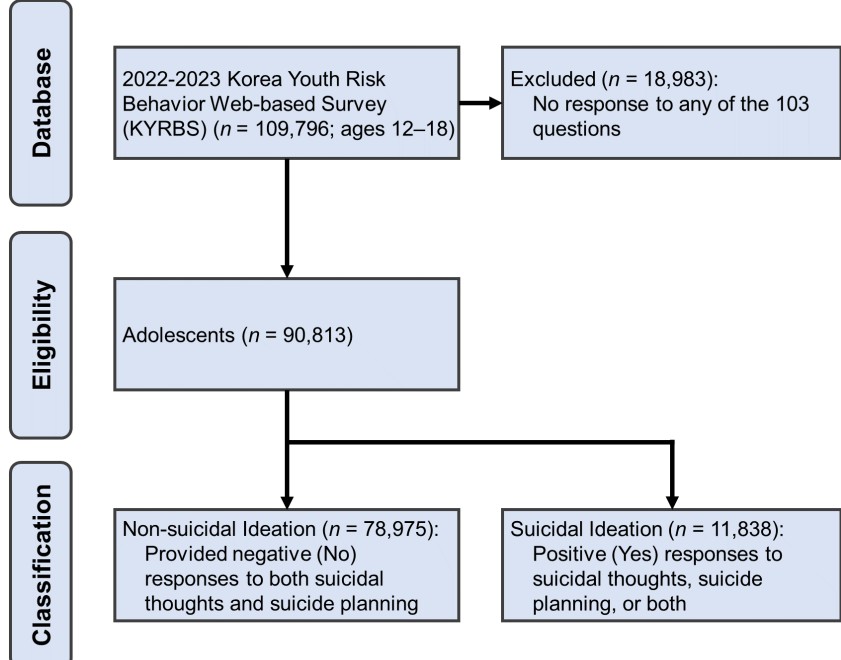

**Fig 1. Strengthening the Reporting of Observational Studies in Epidemiology (STROBE) diagram of eligibility determinations and group classification of participants.**

no) as part of the KYRBS. Participants who answered "yes" were categorized as the suicidal ideation group (n = 11,838), whereas those who answered "no" were categorized as the non-suicidal ideation group (n = 78,975). For the analysis of suicide attempts, participants were further classified according to their responses to the question asking whether they had ever attempted suicide during the past 12 months (yes/no), resulting in 2,191 participants in the suicide attempt group; additionally, based on direct confirmation from the KYRBS administrators, there was no participant overlap between the suicidal ideation group and the suicide attempt subgroup.

Building on the findings of recent systematic reviews and meta-analyses [9,10], we investigated multidimensional risk factors among the 90,813 adolescents included in this study, focusing on their associations with suicidal behaviors, including suicidal ideation and suicide attempts. Specifically, multidimensional risk factors were categorized into four domains: sociodemographic characteristics, physical health, mental health, and health-related behaviors [9,10]. The sociodemographic factors included sex, academic grade, residential area type, academic performance, and family economic status. Physical health factors included physical activity level, sleep quality, and self-reported health status. Mental health factors included stress, anxiety, sadness, and loneliness. Health-related behavioral factors included alcohol consumption and smoking status. A total of 14 factors across these four domains were analyzed. Furthermore, we integrated responses to suicidal behaviors, including suicidal thoughts, planning, and attempts, into statistical analyses and ML models.

## Statistical analysis

To evaluate the effects of these 14 factors on suicidal ideation, we conducted statistical analyses using the IBM SPSS Statistics software (IBM Corp., Armonk, NY, USA). Categorical variables are presented as numbers and percentages, whereas continuous variables are reported as means ± standard deviations. Categorical variables were analyzed using the chi-square test to determine significant differences between the non-suicidal ideation and suicidal ideation groups. For continuous variables, the Shapiro–Wilk test was performed to assess normality. As all continuous variables were not normally distributed, the Mann–Whitney U test was used to compare differences in variables between the non-suicidal ideation and suicidal ideation groups.

To ensure a robust interpretation of the results, effect sizes were calculated to address the possibility that statistically significant differences resulted from a large sample size rather than meaningful effects. Specifically, the Cramer's V value was calculated to measure the strength of the association between categorical variables, with values ranging from 0 (no association) to 1 (perfect association), making it particularly suitable for chi-square analysis [12]. For continuous variables, Cohen's d was calculated to quantify the magnitude of the differences between group means, expressed in standard deviation units, facilitating consistent interpretation across variables [13]. Statistical significance was set at two-sided $p < 0.05$.

## Machine learning framework and optimal feature selection

The ML framework depicted in Fig 2 incorporates several key processes: 1) training and validation of the ML model, 2) identification of the minimal subset of risk factors (i.e., optimal features) associated with suicidal ideation, and 3) validation of the model's performance in predicting suicide attempts. We implemented and executed this ML framework using custom scripts created in MATLAB (MathWorks, Natick, MA, USA).

To address the class imbalance between the non-suicidal ideation group (n = 78,975) and the suicidal ideation group (n = 11,838), we applied the synthetic minority oversampling technique (SMOTE) to the dataset. SMOTE generates synthetic samples for the minority class to achieve a balanced distribution within the training and validation datasets [14]. Our ML framework incorporated bootstrapping and recursive feature elimination, both of which are well-established techniques for enhancing ML performance [15,16]. Considering the sample sizes (i.e., n = 78,975 and n = 11,838) and recommendations from previous studies [17,18], 50 bootstrapping iterations were conducted to ensure robustness in assessing the performance of the ML model across varying data distributions.

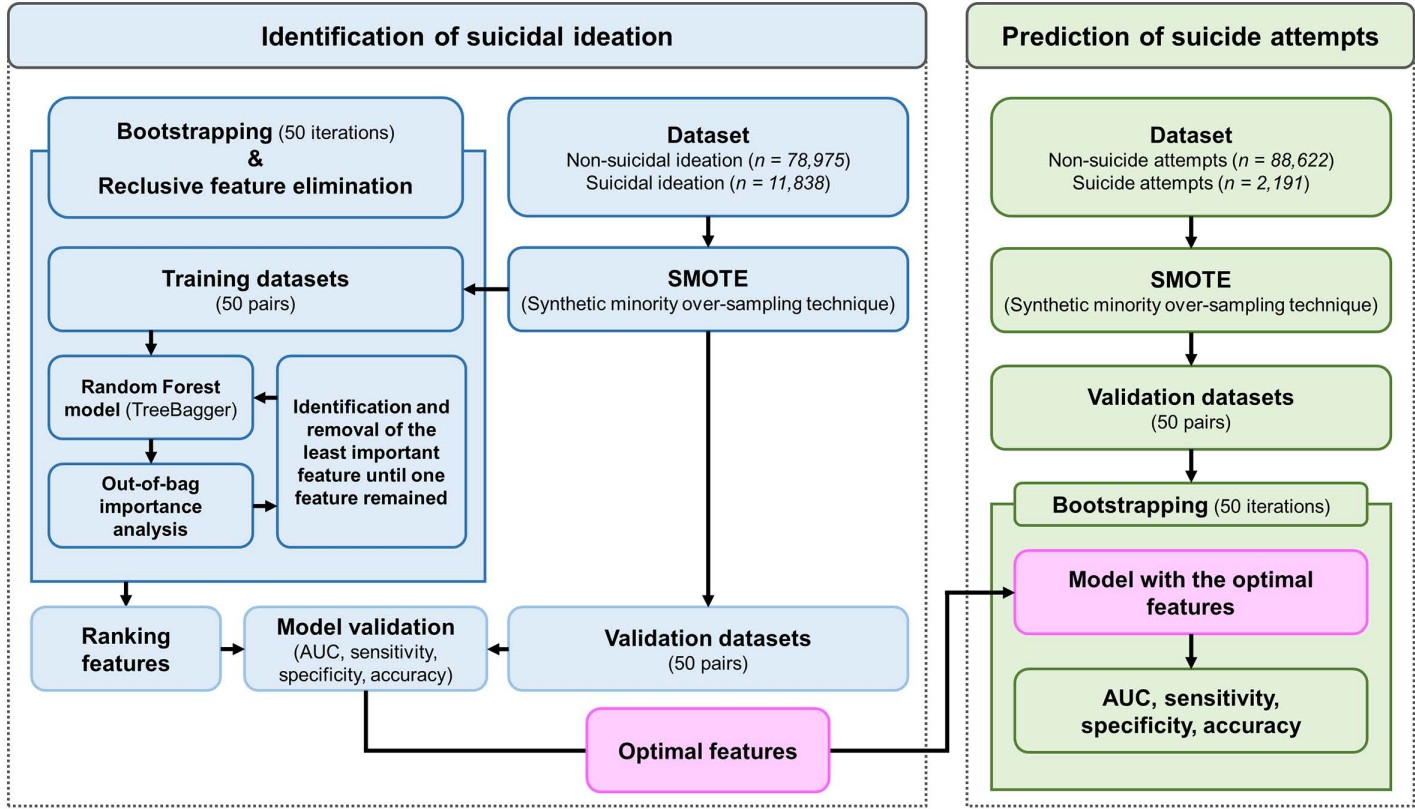

**Fig 2. The machine learning framework is under the curve (AUC), the area under the receiver operating characteristic curve.**

The Random Forest (RF) model was selected based on the evidence from recent systematic reviews and meta-analyses [9,10], which demonstrate its widespread application and high predictive accuracy in identifying and forecasting suicidal behaviors among adolescents. The RF is an ensemble-based ML approach that combines multiple decision trees to produce robust and precise predictions.

In the RF model, the dependent variable was defined as a categorical variable, with 0 representing the non-suicidal ideation group and 1 representing the suicidal ideation group. The independent variables (i.e., features) were selected from the 14 factors analyzed based on their statistical significance and practical relevance [19]. Specifically, factors were included if they met the criterion of statistical significance ($p < 0.05$) and if they demonstrated effect sizes ranging from small to large. This rigorous selection methodology was applied to ensure that the RF model incorporated features with both statistically validated and practically meaningful contributions, thereby enhancing its interpretability and robustness in predicting suicidal ideation.

To evaluate feature importance, a permutation feature importance analysis was conducted using out-of-bag (OOB) data [20]. The predictive accuracy was initially assessed using the original feature values to calculate the baseline OOB error. Subsequently, the values of each feature were randomly permuted within the OOB dataset, and the OOB error was recalculated. The difference between the baseline and permuted OOB errors provided an importance score for each feature. The features with the lowest importance scores were iteratively removed, and this recursive process continued until only one feature remained (i.e., recursive feature elimination).

Following the ranking of features through recursive feature elimination, the RF models were re-evaluated using validation datasets to assess their performance with the reduced feature set. During the validation process, performance

metrics, such as the area under the receiver operating characteristic curve (AUC), sensitivity, specificity, and accuracy, were calculated [15,16,21]. The AUC measured the model's capacity to distinguish between adolescents with and without suicidal ideation, serving as an overall indicator of classification effectiveness. Sensitivity reflected the model's ability to accurately identify cases of suicidal ideation, whereas specificity quantified its precision in correctly classifying non-suicidal ideation. The accuracy represented the overall classification performance of the model across all categories.

The optimal feature set was determined when all performance metrics (AUC, sensitivity, specificity, and accuracy) consistently exceeded 80%, ensuring robust and reliable model performance. Notably, an AUC of > 80% is generally considered excellent [22].

To evaluate the predictive performance of the different ML models using either the optimal feature set or sets that included high-ranking features beyond the optimal set for predicting suicide attempts, the same dataset comprising adolescents with and without suicide attempts (n = 88,622 and n = 2,191, respectively) was used. SMOTE was applied to the dataset to address the class imbalance and generate validation datasets. Model validation was conducted using 50 bootstrapping iterations, during which AUC, sensitivity, specificity, and accuracy were calculated to assess the model's predictive capabilities.

### Ethics statement

The use of the KYRBS data and study protocol were approved by the Institutional Review Board of Yonsei University (IRB No. 1041849–202411-BM-226–01; approval date: November 7, 2024). Owing to the retrospective design of the study using anonymized data, the need to obtain informed consent was waived by the Institutional Review Board of Yonsei University. All methods in this study were performed in accordance with the guidelines and regulations of the Declaration of Helsinki. Data access for research purposes occurred on November 8, 2024, following IRB approval.

## Results

### Multidimensional risk factors contributing to suicidal ideation

Table 1 presents the descriptive statistics and results of the statistical analyses of the multidimensional risk factors associated with suicidal ideation. Among the sociodemographic factors, sex, academic performance, and family economic status demonstrated significant differences between the groups. Female adolescents demonstrated a higher prevalence of suicidal ideation than males. Academic performance and family economic status were poorer in the suicidal ideation group than in the non-suicidal ideation group.

Physical and mental health factors were significantly associated with suicidal ideation. Adolescents in the suicidal ideation group reported significantly poorer sleep quality and lower self-reported health status than those in the non-suicidal ideation group. A significantly larger proportion of adolescents in the suicidal ideation group experienced sadness than those in the non-suicidal ideation group. Additionally, stress levels, anxiety, and loneliness scores were significantly higher in the suicidal ideation group.

Health-related behavioral factors, including alcohol consumption and smoking status, were also significantly more prevalent among adolescents with suicidal ideation. The suicidal ideation group demonstrated a significantly higher prevalence of alcohol consumption and smoking than did the non-suicidal group.

Among the 14 multidimensional risk factors examined, 8 factors (sex, family economic status, sleep quality, self-reported health status, sadness, stress, anxiety, and loneliness) were statistically significant (p < 0.05) and practically relevant (i.e., small to large effect sizes) in their association with suicidal ideation. Consequently, these eight factors were included in the machine learning model.

### Performance of the machine learning model and optimal feature selection

Table 2 presents the rankings of the eight factors included in the ML model and the validation results for suicidal ideation. The results were evaluated using four performance metrics: AUC, sensitivity, specificity, and accuracy. Fig 3A illustrates

**Table 1. Multidimensional risk factors associated with suicidal ideation, including sociodemographic characteristics, physical health, mental health, and health-related behaviors.**

| Variables | Non-suicidal ideation (n = 78,975) | Suicidal ideation (n = 11,838) | p-value | Effect size |
|---|---|---|---|---|
| **Sociodemographic factors** | | | | |
| *Sex (female), n (%) | 37,887 (47.97) | 7,599 (64.19) | < 0.0001 | 0.11 |
| Academic grade (high school), n (%) | 36,679 (46.44) | 4,731 (39.96) | < 0.0001 | 0.04 |
| Residential area type, score (1: big–5: small) | 1.64 ± 0.62 | 1.65 ± 0.60 | 0.208 | 0.01 |
| Academic performance, score (1: good–5: bad) | 2.86 ± 1.14 | 3.04 ± 1.21 | < 0.0001 | 0.15 |
| *Family economic status, score (1: good–5: bad) | 2.57 ± 0.85 | 2.75 ± 0.95 | < 0.0001 | 0.20 |
| **Physical health factors** | | | | |
| Physical activity levels, score (1: none–8: 7 days) | 3.19 ± 2.16 | 3.11 ± 2.15 | < 0.0001 | 0.04 |
| *Sleep quality, score (1: good–5: bad) | 3.18 ± 1.09 | 3.67 ± 1.05 | < 0.0001 | 0.46 |
| *Self-reported health status, score (1: good–5: bad) | 2.19 ± 0.87 | 2.68 ± 0.97 | < 0.0001 | 0.53 |
| **Mental health factors** | | | | |
| *Sadness, n (%) | 15,994 (20.25) | 8,255 (69.73) | < 0.0001 | 0.38 |
| *Stress, score (1: a lot–5: none) | 2.83 ± 0.88 | 1.95 ± 0.81 | < 0.0001 | 1.01 |
| *Anxiety, score (1: minimal–4: severe) | 2.60 ± 0.70 | 3.41 ± 0.69 | < 0.0001 | 1.12 |
| *Loneliness, score (1: none–5: always) | 2.42 ± 1.00 | 3.50 ± 0.96 | < 0.0001 | 1.10 |
| **Health-related behavioral factors** | | | | |
| Alcohol consumption, n (%) | 24,653 (31.22) | 5,105 (43.12) | < 0.0001 | 0.09 |
| Smoking status, n (%) | 5,658 (7.16) | 1,647 (13.91) | < 0.0001 | 0.08 |

The asterisk (*) indicates statistically significant and practically relevant effects

the average values of these performance metrics (AUC, sensitivity, specificity, and accuracy) with their corresponding 95% confidence intervals plotted as a function of the ranked factors. Fig 3B presents the traditional receiver operating characteristic (ROC) curves for the eight cumulative models used to identify suicidal ideation, with all curves overlaid in a single graph (sensitivity vs. 1 – specificity). Each curve displays its respective AUC value, demonstrating that all models achieved strong discriminatory power (AUC > 78.86%) and that predictive performance progressively improved as additional features were incorporated.

The performance of the model stabilized and demonstrated its effectiveness for the four factors (sadness, loneliness, anxiety, and stress). Under these conditions, the model achieved an AUC exceeding 90%, accompanied by a sensitivity, specificity, accuracy, PPV (Positive Predictive Value), and NPV (Negative Predictive Value) exceeding 80%, thereby demonstrating robust predictive capabilities. Consequently, these four factors were determined to be the optimal features (i.e., the minimal subset of risk factors), which are required to maintain high classification performance while enhancing interpretability and computational efficiency.

## Predictive performance of machine learning model for suicide attempts

Table 3 presents the predictive performances of the five ML models for suicide attempts, demonstrating progressive enhancement with the inclusion of additional features. Model 1, which incorporated the optimal features (sadness, loneliness, anxiety, and stress), achieved an AUC of 90.81%, a sensitivity of 84.09%, a specificity of 83.27%, an accuracy of 83.68%, a PPV of 83.41%, and an NPV of 83.95%. When family economic status was added to Model 2, the predictive performance improved, with an AUC of 93.06%, sensitivity of 87.16%, specificity of 84.21%, and accuracy of 85.68%, a PPV of 84.67%, and an NPV of 86.77%.

**Table 2. Ranking of the eight factors included in the machine learning model and validation results of the model using 50 validation datasets.**

| Rank | Factor | AUC (mean) | Sensitivity (mean) | Specificity (mean) | Accuracy (mean) | PPV (mean) | NPV (mean) | Included rank(s) |
|---|---|---|---|---|---|---|---|---|
| 1 | Sadness | 78.76% | 81.52% | 74.48% | 78.00% | 76.16% | 80.13% | 1 |
| 2 | Loneliness | 86.77% | 81.42% | 78.93% | 80.17% | 79.44% | 80.95% | 1–2 |
| 3 | Anxiety | 89.05% | 83.80% | 80.12% | 81.96% | 80.82% | 83.19% | 1–3 |
| 4 | Stress | 90.91% | 83.94% | 83.50% | 83.72% | 83.57% | 83.88% | 1–4 |
| 5 | Family economic status | 93.04% | 86.90% | 84.46% | 85.68% | 84.83% | 86.58% | 1–5 |
| 6 | Sleep quality | 94.73% | 88.97% | 86.80% | 87.88% | 87.08% | 88.73% | 1–6 |
| 7 | Self-reported health status | 96.49% | 92.06% | 89.04% | 90.55% | 89.36% | 91.81% | 1–7 |
| 8 | Sex | 97.22% | 93.51% | 90.06% | 91.79% | 90.39% | 93.28% | 1–8 |

Further enhancement was observed in Model 3 with the inclusion of sleep quality, resulting in an AUC of 94.81%, sensitivity of 89.28%, specificity of 86.76%, and accuracy of 88.02%, a PPV of 87.09%, and an NPV of 89.00%. Model 4, which included self-reported health status, demonstrated an even higher predictive performance, achieving an AUC of 96.55%, sensitivity of 92.08%, specificity of 89.10%, and accuracy of 90.59%, a PPV of 89.41%, and an NPV of 91.48%. Finally, Model 5, which incorporated sex, showed the highest predictive performance. The AUC reached 97.28%, sensitivity of 93.49%, specificity of 90.21%, and accuracy of 91.85%, a PPV of 90.52%, and an NPV of 93.26%.

## Discussion

This study explored the significance of identifying a minimal set of critical risk factors for adolescent suicidal behaviors (i.e., suicidal ideation and suicide attempts) using an ML approach. Consistent with the existing literature, the findings of this study emphasize the multidimensional nature of adolescent suicidal behaviors, which includes sociodemographic characteristics, physical and mental health, and health-related behaviors. Using data from the KYRBS, this study demonstrated the efficacy of ML models in accurately identifying suicidal ideation and predicting suicide attempts. Notably, the ML models identified a minimal subset of risk factors (i.e., optimal features), including sadness, loneliness, anxiety, and stress, which ensured a robust and precise classification of suicidal ideation and the prediction of suicide attempts.

The results of this study highlight the critical role of optimal features as key risk factors for suicidal behaviors among adolescents. These results are consistent with a growing body of evidence emphasizing the psychological dimensions of suicidality in this population, particularly the strong association between elevated stress levels and increased suicide risk [23]. Sadness and loneliness were also identified as the most influential factors, consistent with previous studies indicating that emotional isolation exacerbates vulnerability to suicidal behaviors [24]. These results are particularly significant, given that adolescence represents a developmental period marked by elevated emotional sensitivity and social reliance, during which perceived or actual isolation can profoundly exacerbate psychological distress. Additionally, the role of anxiety is consistent with its established association with cognitive distortions, such as catastrophic thinking and rumination, which have been shown to intensify suicidal behavior [25,26].

In addition to psychological factors, this study identified sex, family economic status, sleep quality, and self-reported health status as significant secondary risk factors for suicidal behaviors. Previous research has consistently indicated that female adolescents exhibit higher rates of suicidal behaviors than their male counterparts. This disparity has been attributed to interrelated factors, including gender differences in emotional expression, coping mechanisms, and heightened exposure to stressors such as interpersonal conflicts and academic pressures [27]. Additionally, hormonal fluctuations during puberty may intensify emotional sensitivity and increase susceptibility to mood disorders, both of which are strongly associated with suicidal behaviors [28]. The association between economic hardship and an elevated risk of suicidal behaviors can be attributed to the cumulative stressors of financial instability, including restricted access to mental

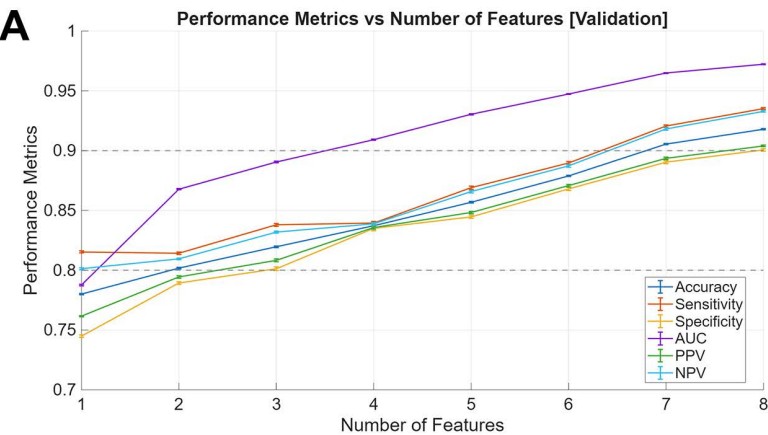

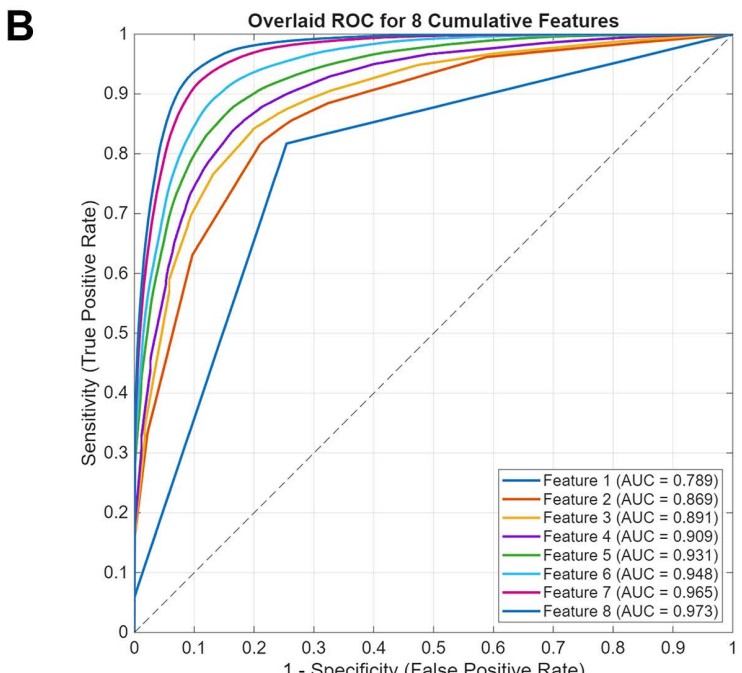

**Fig 3. (A) Performance metrics (accuracy, sensitivity, specificity, and AUC) plotted against the number of included features during model validation, with error bars representing 95% confidence intervals.** AUC: area under the curve. PPV: Positive Predictive Value. NPV: Negative Predictive Value. (B) Overlaid receiver operating characteristic (ROC) curves for the eight cumulative models used to identify suicidal ideation (sensitivity vs. 1 – specificity). Each curve displays its corresponding AUC value, demonstrating progressive improvement in model performance with additional features. AUC: area under the curve.

health resources and increased familial conflict, which collectively foster an environment conducive to psychological distress [29,30]. Additionally, sleep quality was identified as a significant risk factor, corroborating growing evidence linking sleep disturbances to emotional dysregulation and impaired executive function, both of which play pivotal roles in the regulation of suicidal behaviors [31,32]. Disrupted sleep patterns, such as insomnia and irregular sleep schedules, have been associated with diminished neural plasticity and altered emotional processing, thereby exacerbating vulnerability to suicidal behaviors in adolescents [33]. Similarly, the identification of self-reported health status as a significant risk factor

**Table 3. Predictive performance of machine learning models for suicide attempts.**

| Model number | Features | AUC | Sensitivity | Specificity | Accuracy | PPV | NPV |
|---|---|---|---|---|---|---|---|
| 1 | Sadness, loneliness, anxiety, stress | 90.81±0.06% | 84.09±0.13% | 83.27±0.20% | 83.68±0.09% | 83.41±0.16% | 83.95±0.10% |
| 2 | Model 1 features combined with family economic status | 93.06±0.05% | 87.16±0.12% | 84.21±0.17% | 85.68±0.08% | 84.67±0.13% | 86.77±0.10% |
| 3 | Model 2 features combined with sleep quality | 94.81±0.04% | 89.28±0.09% | 86.76±0.15% | 88.02±0.06% | 87.09±0.12% | 89.00±0.08% |
| 4 | Model 3 features combined with self-reported health status | 96.55±0.03% | 92.08±0.08% | 89.10±0.10% | 90.59±0.05% | 89.41±0.09% | 91.84±0.07% |
| 5 | Model 4 features combined with sex | 97.28±0.02% | 93.49±0.08% | 90.21±0.09% | 91.85±0.05% | 90.52±0.07% | 93.26±0.08% |

The reported values for area under the curve (AUC), sensitivity, specificity, accuracy, Positive Predictive Value (PPV), and Negative Predictive Value (NPV) represent the mean values ± their corresponding 95% confidence intervals calculated across validation datasets

highlights the intricate relationship between perceived physical well-being and mental health. Poor self-rated health often coexists with feelings of hopelessness and despair, which are foundational components of suicidality [34].

The robustness of the RF model in this study demonstrates its applicability in suicide risk assessment, consistent with prior findings demonstrating that ensemble methods surpass traditional statistical approaches and other ML models (e.g., logistic regression), when applied to high-dimensional datasets [9]. These findings could potentially advance the methodological framework for suicide risk assessment by validating the use of recursive feature elimination to identify optimal features. An AUC exceeding 90% with only four features suggests the potential to optimize predictive processes for clinical applications without compromising precision. This efficiency meets the growing demand for interpretable and computationally viable ML models in clinical contexts [35,36]. Moreover, recursive feature elimination enhances model interpretability and facilitates efficient integration into resource-constrained healthcare settings, enabling the early detection of high-risk adolescents. By prioritizing critical risk factors and maintaining a robust predictive performance, this approach can enhance decision-making processes in clinical environments with limited resources. The application of SMOTE to address class imbalances further reinforces the generalizability of the study's findings [14]. Combined with rigorous validation via bootstrapping, the SMOTE ensured robust performance metrics (sensitivity, specificity, and AUC) across all validation sets. These methodological advancements align with established best practices for reliable ML systems to predict suicide risk behaviors [37,38].

Although this study provides valuable insights into the classification and prediction of suicidal ideation and suicide attempts among adolescents, two limitations must be acknowledged. First, reliance on self-reported survey data may introduce reporting bias, as adolescents may underreport or exaggerate their experiences due to social desirability or recall inaccuracies. Furthermore, although the KYRBS dataset offers comprehensive coverage of Korean adolescents, its generalizability to other cultural contexts remains uncertain, underscoring the need for further validation using datasets from diverse populations. Additionally, the KYRBS dataset used in this study does not include variables explicitly specifically assessing physical or sexual abuse. The absence of such trauma-related variables may limit the comprehensiveness of the risk factor assessment, as exposure to abuse is a well-documented determinant of suicidal behaviors among adolescents. Future studies incorporating datasets that include abuse- or trauma-related items are worthwhile to provide a more complete understanding of the multidimensional contributors to suicidal behaviors.

Nevertheless, identifying a minimal set of predictive factors has significant practical and clinical implications. Clinicians can allocate resources more effectively by prioritizing interventions for adolescents who exhibit high levels of sadness, loneliness, anxiety, and stress. Moreover, the simplicity of the model enhances its feasibility for integration into school-based mental health programs where early detection can play a pivotal role in preventing the escalation of suicidal

behaviors. The demonstrated accuracy of our model in predicting suicide attempts further underscores its potential utility in preventive interventions, offering a promising framework for addressing adolescent mental health challenges.

## Conclusion

This study emphasizes the critical importance of identifying the minimal risk factors for adolescent suicidal behaviors, including suicidal ideation and suicide attempts, using a rigorous ML approach. Using a nationally representative dataset of Korean adolescents, we developed a robust ML model that demonstrated its capability to accurately identify suicidal ideation and predict suicide attempts. Importantly, the results of this study identify a minimal subset of four risk factors (i.e., sadness, loneliness, anxiety, and stress) that are sufficient to maintain high predictive accuracy while enhancing model interpretability, which can reduce computational overhead and facilitate integration into clinical and school-based mental health programs. Additionally, the inclusion of additional factors, such as sex, family economic status, sleep quality, and self-reported health status, further enhances predictive performance, illustrating the interplay between sociodemographic, lifestyle, and psychological risk factors for suicidal behaviors.

The findings of this study have the potential to offer a scalable and efficient framework for early detection of high-risk adolescents. By focusing on the identified psychological dimensions, mental health professionals can implement precise and resource-efficient intervention strategies. The high accuracy of the ML model suggests its potential as a practical tool across diverse healthcare settings, enabling timely and effective interventions to prevent suicidal behaviors. In addition, these results advocate for the integration of ML-driven frameworks into educational systems to enhance suicide prevention efforts. School-based applications of the proposed model can support routine screenings and facilitate early risk identification, ensuring prompt support for vulnerable adolescents. Moreover, the interpretability of the model equips educators and counselors with actionable insights into critical risk factors, enabling data-driven and informed decision-making.

## Acknowledgments

We are grateful to the nationwide Korea Youth Risk Behavior Web-based Survey (KYRBS) registry.

## Author contributions

**Conceptualization:** Catherine Park, Beom-Chan Lee.

**Data curation:** Catherine Park, Beom-Chan Lee.

**Formal analysis:** Catherine Park, Beom-Chan Lee.

**Funding acquisition:** Catherine Park.

**Investigation:** Catherine Park, Beom-Chan Lee.

**Methodology:** Catherine Park, Beom-Chan Lee.

**Project administration:** Catherine Park, Beom-Chan Lee.

**Resources:** Catherine Park.

**Software:** Catherine Park, Beom-Chan Lee.

**Supervision:** Beom-Chan Lee.

**Validation:** Catherine Park, Beom-Chan Lee.

**Visualization:** Catherine Park, Beom-Chan Lee.

**Writing – original draft:** Catherine Park, Beom-Chan Lee.

**Writing – review & editing:** Catherine Park, Beom-Chan Lee.

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
