## [Decision Letter · Decision Letter 0]

11 Nov 2025

Dear Dr. Lee,

Thank you for submitting your manuscript to PLOS ONE. After careful consideration, we feel that it has merit but does not fully meet PLOS ONE’s publication criteria as it currently stands. Therefore, we invite you to submit a revised version of the manuscript that addresses the points raised during the review process.

We look forward to receiving your revised manuscript.

Kind regards,

Vincenzo De Luca

Academic Editor

PLOS ONE

Journal Requirements:

https://journals.plos.org/plosone/s/file?id=ba62/PLOSOne_formatting_sample_title_authors_affiliations.pdf....

“This work was supported by the MSIT (Ministry of Science and ICT), Korea, under the National Program in Medical AI Semiconductor supervised by the IITP (Institute of Information & Communications Technology Planning & Evaluation) [grant number 2024-0-0096]. The funding sources had no role in the study design, methods, data collection and analysis, and submission of the results.”

4. Please be informed that funding information should not appear in the Acknowledgments section or other areas of your manuscript. We will only publish funding information present in the Funding Statement section of the online submission form. Please remove any funding-related text from the manuscript.

Reviewers' comments:

Reviewer's Responses to Questions

**Comments to the Author**

1. Is the manuscript technically sound, and do the data support the conclusions?

Reviewer #1: Yes

2. Has the statistical analysis been performed appropriately and rigorously?

Reviewer #1: Yes

3. Have the authors made all data underlying the findings in their manuscript fully available?

Reviewer #1: Yes

4. Is the manuscript presented in an intelligible fashion and written in standard English?

Reviewer #1: Yes

Reviewer #1: The rationale for this study was the fact that “suicide is the 3rd leading cause of death for adolescents world-wide.” It used data from the Korea Youth Risk Behavior Web-based Survey (2022–2023, Korea Disease Control and Prevention Agency, KDCA), including 90,217 adolescents aged 12–18 years. Suicidal: yes=11,838, no=78,975 (6.67%); attempts 2,191.

Data were analyzed via machine learning (Random Forest, RF) model, which they note is superior to meta-analysis. They used multidimensional risk factors (psychological, sociological, physical illness, demographic ) spanning sociodemographic, physical and mental health, and behavioral domains. Sensitivity and specificity were assessed using the AUC method. “Sadness, loneliness, anxiety, and stress were identified as optimal features, achieving strong results in identifying suicidal ideation and predicting suicide attempts” == AUC 92%, Sens 94%, Spec 90%.

Cramer’s V was used to define effect size of categoricals, Cohen’s-d for effect size of continuous variables.

[how was suicidal vs. non-suicidal assessed?]

Of note, statistical risk factors by effect size were: Anxiety > Loneliness > Stress > Sleep > Sadness. These all had AUC >78%. While alcohol and smoking had very low effect size. 8 factors (sex, family economic status, sleep quality, self-reported health status, sadness, stress, anxiety, and loneliness) were statistically significant (p < 0.05) and relevant to clinical practice. Five statistical models comprised various combinations of these factors, with the most predictive (5) having sens=93.5%, spec= 90.3%

One wonders whether physical or sexual abuse was asked about in this large survey, and if so, what effect it would have on the results. Otherwise, the stated limitations are reasonable.

Fig. 3 displays factors that were internal to the machine learning method but are presented in a way that is unfamiliar to readers accustomed to reading AUC curves. I suggest that the authors either add a traditional AUC curve (Sensitivity vs. 1 – Specificity) for each of the 8 factors in their analysis, with all 8 AUCs overlaid on the same graph, or replace Fig. 3 with the latter AUC curve.

Overall, this paper is an excellent guide for those working with suicidal adolescents

.

Reviewer #1: No

---

## [Author Response · Author response to Decision Letter 1]

17 Nov 2025

Reviewer 1’s comments

The rationale for this study was the fact that “suicide is the 3rd leading cause of death for adolescents world-wide.” It used data from the Korea Youth Risk Behavior Web-based Survey (2022–2023, Korea Disease Control and Prevention Agency, KDCA), including 90,217 adolescents aged 12–18 years. Suicidal: yes=11,838, no=78,975 (6.67%); attempts 2,191.

Data were analyzed via machine learning (Random Forest, RF) model, which they note is superior to meta-analysis. They used multidimensional risk factors (psychological, sociological, physical illness, demographic) spanning sociodemographic, physical and mental health, and behavioral domains. Sensitivity and specificity were assessed using the AUC method. “Sadness, loneliness, anxiety, and stress were identified as optimal features, achieving strong results in identifying suicidal ideation and predicting suicide attempts” == AUC 92%, Sens 94%, Spec 90%.

Cramer’s V was used to define effect size of categoricals, Cohen’s-d for effect size of continuous variables.

[how was suicidal vs. non-suicidal assessed?]

Of note, statistical risk factors by effect size were: Anxiety > Loneliness > Stress > Sleep > Sadness. These all had AUC >78%. While alcohol and smoking had very low effect size. 8 factors (sex, family economic status, sleep quality, self-reported health status, sadness, stress, anxiety, and loneliness) were statistically significant (p < 0.05) and relevant to clinical practice. Five statistical models comprised various combinations of these factors, with the most predictive (5) having sens=93.5%, spec= 90.3%

One wonders whether physical or sexual abuse was asked about in this large survey, and if so, what effect it would have on the results. Otherwise, the stated limitations are reasonable.

Fig. 3 displays factors that were internal to the machine learning method but are presented in a way that is unfamiliar to readers accustomed to reading AUC curves. I suggest that the authors either add a traditional AUC curve (Sensitivity vs. 1 – Specificity) for each of the 8 factors in their analysis, with all 8 AUCs overlaid on the same graph, or replace Fig. 3 with the latter AUC curve.

Overall, this paper is an excellent guide for those working with suicidal adolescents

Response:

We sincerely appreciate the reviewer’s thorough evaluation and constructive feedback. Below, we have restated each comment (in bold) and provided detailed responses (in regular text). All corresponding revisions in the manuscript have been incorporated and are highlighted in grey.

Comments:

1. How was suicidal vs. non-suicidal assessed?

Response:

We thank the reviewer for pointing out the need for clearer clarification of the classification criteria.

In the revised manuscript, we have now explicitly described how suicidal ideation and suicide attempts were defined based on the KYRBS survey questions. Specifically, suicidal ideation and non-suicidal groups were classified according to responses to the following KYRBS items (e.g., During the past 12 months, have you ever seriously thought about committing suicide?; During the past 12 months, have you ever made a specific plan to commit suicide?; and During the past 12 months, have you ever attempted suicide?).

Participants who responded “Yes” to either suicidal thoughts or suicide planning were categorized into the suicidal ideation group, whereas those who answered “No” to both questions were classified as the non-suicidal group. For the analysis of suicide attempts, participants who responded “Yes” to the suicide attempt question were classified into the suicide attempt group, and those who answered “No” were classified as non-attempt. This clarification has been added to the Data Source, Participant Selection, and Multidimensional Risk Factors section.

2. One wonders whether physical or sexual abuse was asked about in this large survey, and if so, what effect it would have on the results.

Response:

We examined the available variables from the KYRBS dataset and found no items specifically assessing physical or sexual abuse, such as sexual victimisation or non-partner violence. Because these trauma-related exposures are known risk factors for suicidal ideation and suicide attempts, the absence of such items in the dataset may limit the comprehensiveness of our risk-factor assessment. We have therefore added a statement in the Limitations section to that effect and recommend that future research incorporate datasets which include trauma or abuse-related items.

3. Fig. 3 displays factors that were internal to the machine learning method but are presented in a way that is unfamiliar to readers accustomed to reading AUC curves. I suggest that the authors either add a traditional AUC curve (Sensitivity vs. 1 – Specificity) for each of the 8 factors in their analysis, with all 8 AUCs overlaid on the same graph, or replace Fig. 3 with the latter AUC curve.

Response:

We greatly appreciate this valuable suggestion. Following the reviewer’s recommendation, we have revised Figure 3 to include a traditional receiver operating characteristic (ROC) curve presentation. The updated figure now shows the overlaid ROC curves (Sensitivity vs. 1 – Specificity) for each model. We have also updated the corresponding Results subsection and Figure 3 legend to reflect these changes.

---

## [Decision Letter · Decision Letter 1]

23 Feb 2026

Dear Dr. Lee,

Thank you for submitting your manuscript to PLOS ONE. After careful consideration, we feel that it has merit but does not fully meet PLOS ONE’s publication criteria as it currently stands. Therefore, we invite you to submit a revised version of the manuscript that addresses the points raised during the review process.

We look forward to receiving your revised manuscript.

Kind regards,

Vincenzo De Luca

Academic Editor

PLOS One

Journal Requirements:

Reviewers' comments:

Reviewer's Responses to Questions

**Comments to the Author**

Reviewer #1: All comments have been addressed

Reviewer #2: All comments have been addressed

Reviewer #3: (No Response)

2. Is the manuscript technically sound, and do the data support the conclusions?

Reviewer #1: Yes

Reviewer #2: Yes

Reviewer #3: Partly

3. Has the statistical analysis been performed appropriately and rigorously?

Reviewer #1: Yes

Reviewer #2: Yes

Reviewer #3: No

4. Have the authors made all data underlying the findings in their manuscript fully available?

Reviewer #1: Yes

Reviewer #2: Yes

Reviewer #3: Yes

5. Is the manuscript presented in an intelligible fashion and written in standard English?

Reviewer #1: Yes

Reviewer #2: Yes

Reviewer #3: Yes

Reviewer #1: All of my questions were answered to my satisfaction. Especially, the multiple AUC graph is now fully understandable, and quite dramatic in its accuracy.

Reviewer #2: The manuscript addresses an important and timely public health issue, namely adolescent suicidal ideation and suicide attempts, and applies a machine learning-optimized approach to identify minimal risk factors. The topic is highly relevant, and the overall structure of the paper is clear and logically organized.

The objectives of the study are well stated, and the use of machine learning methods appears appropriate for the research question. The manuscript is generally well written and understandable, and the results are presented in a coherent manner.

I did not identify any major concerns related to the overall approach, ethical considerations, or presentation that would preclude publication. Minor improvements may be possible in terms of clarity or detail in some sections, but these do not affect the main conclusions of the study.

Overall, the manuscript makes a useful contribution to the literature on adolescent mental health and suicide prevention.

Reviewer #3: The manuscript presents an original investigation aiming to develop and validate a machine learning (ML) model to identify minimal risk factors associated with adolescent suicidal ideation and suicide attempts. Data from a large nationwide online survey (Korea Youth Risk Behavior Web-based Survey, 2022–2023) were used for the analyses. The survey assesses four domains of risk factors: sociodemographic, physical health, mental health, and behavioral domains. The results suggest that the ML model performed well in identifying optimal features associated with suicidal ideation. Subsequently, these selected optimal features, along with additional factors, were used to assess the prediction of suicide attempts. The authors conclude that ML-driven approaches may have potential clinical usefulness in improving adolescent mental health outcomes.

Given the high public health importance of suicide in adolescent populations, the topic is highly relevant. The English language is clear overall, and the structure of the manuscript is appropriate.

However, several methodological questions arose during the review of the manuscript. The comments below are intended as constructive feedback to help strengthen this important and valuable work.

1. Sample characteristics and reporting

Questions arose regarding the analysed sample:

1. The total number of analysed subjects differs across sections of the manuscript (Abstract, ‘Data source, participant selection, and multidimensional risk factors’ section in lines 96–111, and Figure 1; N=90,813 vs. N=90,217). This discrepancy should be clarified.

2. It remains unclear whether respondents could overlap between the 2022 and 2023 KYRBS cohorts (e.g., whether the same individual may have participated in both survey years).

3. Lines 109–111: “For the analysis of suicide attempts, participants were further classified according to their responses to the question asking whether they had ever attempted suicide during the past 12 months (yes/no).”

The total number of suicide attempts (n = 2,191) should also be reported here for clarity (not only later in lines 194–195). Additionally, it would be helpful to report the degree of overlap between this subgroup and the suicidal ideation group.

2. Terminology

I would suggest reconsidering the use of the umbrella term “suicidal behaviors” to describe suicidal ideation and suicide attempts together (e.g., lines 331–332). As the manuscript highlights potential clinical applications of ML-driven approaches, it is important to note that suicidal ideation and suicide attempts are considered distinct constructs with distinct predictors in clinical practice (for example see Klonsky & May, 2013), although they may share some predictors. Clear differentiation may improve conceptual clarity.

3. Methodological consideration regarding feature selection

This conceptual distinction also raises a methodological question: If suicidal ideation and suicide attempts have (partially) different predictors, it may be theoretically problematic to use optimal features derived from a model trained on suicidal ideation to predict suicide attempts. To my knowledge, feature importance in machine learning models is inherently outcome-specific. Therefore, features selected using a model optimized for suicidal ideation may not generalize to suicide attempt prediction. Re-deriving the predictive model using suicide attempt as the target outcome could strengthen the methodological validity of the analysis.

4. Clinical interpretability of model performance

Regarding potential clinical usefulness, the manuscript demonstrates strong model performance primarily using AUC, sensitivity, and specificity metrics. However, for clinical applicability, additional performance measures may be more informative, particularly positive predictive value (PPV) and negative predictive value (NPV). This is especially relevant given the relatively low prevalence of suicide attempts in both the general adolescent population and the analysed dataset.

Overall recommendation

Overall, the study has the potential to make a meaningful contribution to the scientific literature. However, in my opinion, the manuscript would benefit from further methodological clarification and potential additional analyses addressing the concerns outlined above before reconsideration for publication.

.

Reviewer #1: No

Reviewer #2: No

Reviewer #3: No

---

## [Author Response · Author response to Decision Letter 2]

26 Feb 2026

Reviewer 1’s comments

All of my questions were answered to my satisfaction. Especially, the multiple AUC graph is now fully understandable, and quite dramatic in its accuracy.

Response:

Thank you for your positive and constructive feedback. We are pleased to hear that all of your questions were addressed to your satisfaction. In particular, we sincerely appreciate your comment that the multiple AUC figure is now fully understandable and that the accuracy is presented in a compelling manner. We have retained the revised figure and its accompanying explanation to ensure clarity and transparency for readers in the final manuscript.

Reviewer 2’s comments

The manuscript addresses an important and timely public health issue, namely adolescent suicidal ideation and suicide attempts, and applies a machine learning-optimized approach to identify minimal risk factors. The topic is highly relevant, and the overall structure of the paper is clear and logically organized.

The objectives of the study are well stated, and the use of machine learning methods appears appropriate for the research question. The manuscript is generally well written and understandable, and the results are presented in a coherent manner.

I did not identify any major concerns related to the overall approach, ethical considerations, or presentation that would preclude publication. Minor improvements may be possible in terms of clarity or detail in some sections, but these do not affect the main conclusions of the study.

Overall, the manuscript makes a useful contribution to the literature on adolescent mental health and suicide prevention.

Response:

Thank you for your thoughtful and encouraging comments on the significance, clarity, and overall rigor of our manuscript. We appreciate your assessment that the objectives are well stated, the machine learning approach is appropriate, and the results are coherently presented. Following your suggestion and incorporating the other reviewers’ comments and suggestions, we revised the manuscript to improve clarity and add methodological and reporting details where needed, without altering the main conclusions. We are grateful for your consideration of our work.

Reviewer 3’s comments

The manuscript presents an original investigation aiming to develop and validate a machine learning (ML) model to identify minimal risk factors associated with adolescent suicidal ideation and suicide attempts. Data from a large nationwide online survey (Korea Youth Risk Behavior Web-based Survey, 2022–2023) were used for the analyses. The survey assesses four domains of risk factors: sociodemographic, physical health, mental health, and behavioral domains. The results suggest that the ML model performed well in identifying optimal features associated with suicidal ideation. Subsequently, these selected optimal features, along with additional factors, were used to assess the prediction of suicide attempts. The authors conclude that ML-driven approaches may have potential clinical usefulness in improving adolescent mental health outcomes. Given the high public health importance of suicide in adolescent populations, the topic is highly relevant. The English language is clear overall, and the structure of the manuscript is appropriate. However, several methodological questions arose during the review of the manuscript. The comments below are intended as constructive feedback to help strengthen this important and valuable work.

Specific Comments:

1. Sample characteristics and reporting

Questions arose regarding the analysed sample:

1-1. The total number of analysed subjects differs across sections of the manuscript (Abstract, ‘Data source, participant selection, and multidimensional risk factors’ section in lines 96–111, and Figure 1; N=90,813 vs. N=90,217). This discrepancy should be clarified

Response:

Thank you for noting this discrepancy. We confirm that N = 90,813 (as shown in Figure 1) is the correct total, and the inconsistency was due to a manual transcription error during manuscript preparation. We have revised the Abstract and the “Data source, participant selection, and multidimensional risk factors” section to consistently reflect N = 90,813 throughout the manuscript. In addition to this correction, all other revisions addressing your comments have been highlighted in yellow for your convenience.

1-2. It remains unclear whether respondents could overlap between the 2022 and 2023 KYRBS cohorts (e.g., whether the same individual may have participated in both survey years).

Response:

We thank the reviewer for pointing this out. For clarification, we contacted the KYRBS administrators directly and confirmed that the 2022 and 2023 KYRBS were conducted as independent, nationally representative cross-sectional surveys (i.e., point-in-time assessments of the population with no longitudinal follow-up) using random sampling of schools and students in South Korea. Based on this survey design, they confirmed that there was no respondent overlap between the 2022 and 2023 survey years. We have added this clarification to the manuscript in the “Data source, participant selection, and multidimensional risk factors” section as follows:

“The 2022 and 2023 KYRBS were conducted as independent, nationally representative cross-sectional surveys using random sampling of schools and students in South Korea; based on direct confirmation from the KYRBS administrators, there was no respondent overlap between the two survey years.”

1-3. Lines 109–111: “For the analysis of suicide attempts, participants were further classified according to their responses to the question asking whether they had ever attempted suicide during the past 12 months (yes/no).” The total number of suicide attempts (n = 2,191) should also be reported here for clarity (not only later in lines 194–195). Additionally, it would be helpful to report the degree of overlap between this subgroup and the suicidal ideation group.

Response:

We appreciate your suggestion. We have revised the text in the “Data source, participant selection, and multidimensional risk factors” section to explicitly report the total number of suicide attempts (n = 2,191) for clarity. In addition, consistent with our response above, we contacted the KYRBS administrators directly and confirmed that there is no participant overlap between the suicidal ideation group and the suicide attempt subgroup, and we have added this clarification to the manuscript. Specifically, we revised the sentence as follows:

“For the analysis of suicide attempts, participants were further classified according to their responses to the question asking whether they had ever attempted suicide during the past 12 months (yes/no), resulting in 2,191 participants in the suicide attempt group; additionally, based on direct confirmation from the KYRBS administrators, there was no participant overlap between the suicidal ideation group and the suicide attempt subgroup.”

2. Terminology: I would suggest reconsidering the use of the umbrella term “suicidal behaviors” to describe suicidal ideation and suicide attempts together (e.g., lines 331–332). As the manuscript highlights potential clinical applications of ML-driven approaches, it is important to note that suicidal ideation and suicide attempts are considered distinct constructs with distinct predictors in clinical practice (for example see Klonsky & May, 2013), although they may share some predictors. Clear differentiation may improve conceptual clarity.

Response:

Thank you for this important conceptual clarification. We agree that suicidal ideation and suicide attempts are related but clinically distinct outcomes that may involve partially different predictors. In response, we revised the manuscript to improve conceptual clarity by explicitly distinguishing these two outcomes in the Discussion and by limiting the use of the term “suicidal behaviors” to a broader descriptive context in the Conclusion, where it is clearly specified as encompassing both suicidal ideation and suicide attempts. We believe this revision better aligns the manuscript with clinical terminology and interpretation.

3. Methodological consideration regarding feature selection: This conceptual distinction also raises a methodological question: If suicidal ideation and suicide attempts have (partially) different predictors, it may be theoretically problematic to use optimal features derived from a model trained on suicidal ideation to predict suicide attempts. To my knowledge, feature importance in machine learning models is inherently outcome-specific. Therefore, features selected using a model optimized for suicidal ideation may not generalize to suicide attempt prediction. Re-deriving the predictive model using suicide attempt as the target outcome could strengthen the methodological validity of the analysis.

Response:

Thank you for this important methodological comment. We agree that suicidal ideation and suicide attempts are distinct outcomes, but we respectfully note that our study was not designed to identify an attempt-specific feature set via outcome-specific feature selection. Rather, the explicit aim of the study was to identify a minimal, clinically actionable core set of risk factors and then test whether this minimally sufficient set could support prediction across both suicidal ideation and suicide attempts. This is also the framework described in the Methods, where optimal features were first identified for suicidal ideation and subsequently evaluated for suicide-attempt prediction. Importantly, the results, reported in Table 3, empirically support this choice: the ideation-derived four-feature model already showed meaningful performance for suicide attempts, and performance improved progressively as additional factors were added in Models 2–5. Accordingly, re-deriving a separate attempt-specific model would address a different objective from that of the present study, rather than correct a flaw in the current design.

4. Clinical interpretability of model performance: Regarding potential clinical usefulness, the manuscript demonstrates strong model performance primarily using AUC, sensitivity, and specificity metrics. However, for clinical applicability, additional performance measures may be more informative, particularly positive predictive value (PPV) and negative predictive value (NPV). This is especially relevant given the relatively low prevalence of suicide attempts in both the general adolescent population and the analysed dataset.

Response:

Thank you for this important suggestion. We agree that positive predictive value (PPV) and negative predictive value (NPV) are important additional metrics for evaluating the potential clinical applicability of our models, particularly given the relatively low prevalence of suicide attempts in the analyzed dataset. In response, we reran the machine learning analyses and updated Figure 1, Tables 2 and 3, as well as the corresponding results reported in the Results section and the Abstract. It is worth noting that, because the updated analyses were conducted using the same overall framework as in the previous version, including SMOTE and bootstrapping, some performance values in the tables changed slightly; however, these changes did not lead to any significant differences in the main findings or conclusions of our Revision 1 manuscript.

Overall recommendation: Overall, the study has the potential to make a meaningful contribution to the scientific literature. However, in my opinion, the manuscript would benefit from further methodological clarification and potential additional analyses addressing the concerns outlined above before reconsideration for publication.

Response:

Thank you for your thoughtful and constructive overall assessment. We sincerely appreciate your recognition of the potential contribution of this study to the literature on adolescent mental health and suicide prevention. In response to your comments, we have carefully revised the manuscript to provide further methodological clarification and additional detail where needed, and we also conducted additional analyses to strengthen the clinical interpretability of the findings. We believe that these revisions have substantially improved the clarity, rigor, and transparency of the manuscript, while the main findings and conclusions remain unchanged.

---

## [Decision Letter · Decision Letter 2]

16 Mar 2026

Identifying minimal risk factors for adolescent suicidal ideation and suicide attempts: A machine learning-optimized approach

PONE-D-25-47059R2

Dear Dr. Lee,

We’re pleased to inform you that your manuscript has been judged scientifically suitable for publication and will be formally accepted for publication once it meets all outstanding technical requirements.

Kind regards,

Vincenzo De Luca

Academic Editor

PLOS One

Additional Editor Comments (optional):

Reviewers' comments:

Reviewer's Responses to Questions

**Comments to the Author**

Reviewer #3: All comments have been addressed

2. Is the manuscript technically sound, and do the data support the conclusions?

Reviewer #3: Yes

3. Has the statistical analysis been performed appropriately and rigorously?

Reviewer #3: Yes

4. Have the authors made all data underlying the findings in their manuscript fully available?

Reviewer #3: Yes

5. Is the manuscript presented in an intelligible fashion and written in standard English?

Reviewer #3: Yes

Reviewer #3: All of my previous questions have been satisfactorily addressed.

In the current version of the manuscript, the sample size is consistently reported across the different sections of the text. The concern regarding a potential overlap between the investigated cohorts from 2022 and 2023 has been clarified. Similarly, the possible overlap between the suicide attempt and suicidal ideation groups. Consequently, the revised manuscript provides a clearer description of the study cohort.

The theoretical issue raised during the previous review round has also been clarified: the Authors explain that the KYRBS survey was not originally designed to differentiate between suicide attempts and suicidal ideation. In addition, the terminology in the text has been improved: the umbrella term “suicidal behaviors” used in the previous manuscript version has been revised.

The methodological question raised in the previous review round has been satisfactorily addressed. The positive predictive value (PPV) and negative predictive value (NPV) have now been incorporated into the presentation of the results.

Overall, the revised manuscript makes a valuable contribution to the scientific literature

.

Reviewer #3: No

---

## [Editor Report · Acceptance letter]

PONE-D-25-47059R2

PLOS One

Dear Dr. Lee,

I'm pleased to inform you that your manuscript has been deemed suitable for publication in PLOS One. Congratulations! Your manuscript is now being handed over to our production team.

Kind regards,

on behalf of

Dr. Vincenzo De Luca

Academic Editor

PLOS One